# Development and Validation of 2-Azaspiro [4,5] Decan-3-One (Impurity A) in Gabapentin Determination Method Using qNMR Spectroscopy

**DOI:** 10.3390/molecules26061656

**Published:** 2021-03-16

**Authors:** Nataliya E. Kuz’mina, Sergey V. Moiseev, Mikhail D. Khorolskiy, Anna I. Lutceva

**Affiliations:** 1“Scientific Centre for Expert Evaluation of Medicinal Products” of the Ministry of Health of the Russian Federation, Federal State Budgetary Institution, 8/2 Petrovsky Blvd, 127051 Moscow, Russia; MoiseevSV@expmed.ru (S.V.M.); mkhorolski@gmail.com (M.D.K.); Lutceva@expmed.ru (A.I.L.); 2Department of Pharmaceutical and Toxicological Chemistry Named by A.P. Arzamastsev, I.M. Sechenov First Moscow State Medical University (Sechenov University), 8, Bldg. 2 St. Trubetskaya, 119991 Moscow, Russia

**Keywords:** gabapentin, impurity A, validation, limit of the quantitation, linearity, accuracy, repeatability, precision, specificity, robustness, qNMR, HPLC

## Abstract

The authors developed a ^1^H qNMR test procedure for identification and quantification of impurity A present in gabapentin active pharmaceutical ingredient (API) and gabapentin products. The validation studies helped to determine the limit of quantitation and assess linearity, accuracy, repeatability, intermediate precision, specificity, and robustness of the procedure. Spike-and-recovery assays were used to calculate standard deviations, coefficients of variation, confidence intervals, bias, Fisher’s *F* test, and Student’s *t*-test for assay results. The obtained statistical values satisfy the acceptance criteria for the validation parameters. The authors compared the results of impurity A quantification in gabapentin APIs and capsules by using the ^1^H qNMR and HPLC test methods.

## 1. Introduction

Gabapentin (2-[1-(aminomethyl) cyclohexyl] acetic acid) is a synthetic and non-benzodiazepine analogue of γ-aminobutyric acid. Gabapentin (GP) is usually used for epilepsy, symptoms of peripheral neuropathic pain, postherpetic neuralgia, diabetic peripheral neuropathy, acute alcohol withdrawal syndrome, and multiple sclerosis treatment [1,2,3]. Through intramolecular cyclization in solution, GP can form impurity A (ImpA)-2-azaspiro [4,5] decan-3-one, which is a European and American pharmacopoeias classification [4]. ImpA formation from crystalline GP rate depends on its polymorphic modification, temperature, moisture, shredding rate, and presence of some excipients [5,6,7]. Temperature and pH of medium can influence the GP cyclization. Since ImpA demonstrate some toxicity rate (LD_50_ = 300 mg/kg, white mice [5]), its content should be measured in GP drugs and substances. Identification and quantification of ImpA in GP during the pharmacopoeial analysis are carried out using the HPLC method [8]. HPLC is a highly sensitive and selective method, but the results of HPLC measurements are relative and indirect by nature. HPLC determination of ImpA requires generation of a calibration curve using a pharmacopoeial reference standard for ImpA (which accounts for the relative nature of measurements). The measurement by the HPLC method has a combined uncertainty (which accounts for the indirect nature of measurements). Sources of the total standard uncertainty are the peak area measurement in the chromatogram, the test and standard samples weighing, and solvent volumes measurement. Therefore, it would be practical to use an absolute and direct method, for example, qNMR for ImpA quantification. Absolute methods of quantitative analysis are based on known functional relationships and do not require the generation of a calibration curve using a reference standard. qNMR is considered as an absolute method for measuring the molar ratio of the analytes in a test sample, as well as the weight content of one component relative to another component, because the functional relationships between the analytes and the measurands (integrated intensities) are well-known: the molar ratio of the components in a mixture is equal to the ratio of the normalized integrated intensities of the signals of these components. qNMR quantification of an impurity relative to the main component is considered a direct method because of the direct measurement of the ratio of integrated intensities of the main component and impurity signals. Uncertainty of the test result relies only on the uncertainty of the integral intensities ratio measurement [9,10]. The aim of this article is to develop and validate an identification and quantification method of ImpA determination in GP drugs and APIs.

## 2. Results and Discussion

### 2.1. Specificity

GP and ImpA have a similar structure (Figure 1).

Although the structures of GP and ImpA are similar, signal overlap on the ^1^H spectrum can only observed be in the cyclohexane fragment range (1.25–1.70 ppm). Methylene group signals are differentiated: CH_2_-N δ = 3.02 ppm (GP) and 3.24 ppm (ImpA); CH_2_-C=O δ = 2.45 ppm (GP) and 2.28 ppm (ImpA). It should be noted that signals of the ImpA do not overlap with ^13^C satellites of GP signals (Figure 2).

GP drugs of different manufacturers have nonequal set of excipitents in their content. For example, capsules I (300 mg GP dose) have calcium hydrogen phosphate dihydrate, potato starch, magnesium stearate, and PEG in their content. The content of capsules II with the same active substance dose includes lactose monohydrate, corn starch, talc, and magnesium stearate. It should be noted that signals of water-soluble excipients lie outside of the methylene GP and ImpA protons range. They do not prevent ImpA identification and quantification, as can be seen in Figure 3 and Figure 4.

In this way, signal 2.28 and 3.24 ppm are characteristic signals of ImpA. These signals help to identify the impurity presence in GP drugs and substances.

In the qNMR spectroscopy, the mass of the analyte A can be determined from a known mass of the analyte B [9]:(1)$$m_{A}=\frac{I_{A}}{I_{B}}\times \frac{N_{B}}{N_{A}}\times \frac{M_{A}}{M_{B}}\times m_{B}\times P$$
where m_A_ and m_B_ are the mass of the analytes A and B;M_A_ and M_B_ are the molar mass of the analytes A and B;N_A_ and N_B_ are the number of nuclei generating the corresponding signal;P—purity of the analyte B.

Therefore, the ImpA content in the test sample (m_ImpA_) and its weight % (w %) relative to GP can be determined using the following formulas:m_ImpA_ = 0.901(I_ImpA_/I_GP_) m_GP_; w % = 0.901(I_ImpA_/I_GP_)100(2)
where 0.901 is the relation of ImpA and GP molar masses;I_ImpA_ is the integral intensity of any characteristic ImpA signals (2.28; 3.24 ppm) or their mean;I_GP_ is the integral intensity of any characteristic GP signals (2.45; 3.02 ppm) or their mean;m_GP_ is the GP content in the test sample.

### 2.2. Limit of Quantification

In the experimental conditions of qNMR, the limit of quantitation (LOQ) of ImpA is 10 µg/mL (0.025 weight % relative to GP content). At this concentration of ImpA, the signal-to-noise ratio is 10.3.

### 2.3. Linearity and the Analytical Range

The analytical range of the development method is 10–253 µg/mL or 0.025–0.63 weight % relative to GP. It corresponds to 0.25% from nominal content of ImpA in GP APIs (0.1 w %) and 158% from nominal content of ImpA in the GP drug (0.4 w %). In this range were identified validation characteristics of method. The integral intensity of signals δ 3.24 ppm (ImpA) and 3.02 ppm (GP) is used in quantification measurements. Results of linearity evaluation are shown in Table 1.

Based on data shown in Table 1, we built a dependency graph of I_ImpA_ from ImpA content in model mixtures (Figure 5). Statistical characteristics of the established linear regression are shown in Table 2.

As follows from the data in Table 2, the procedure meets all linearity requirements: the correlation coefficient r ≥ 0.990, and the value *a* does not exceed its confidence interval.

### 2.4. Accuracy

The results of accuracy evaluation are given in Table 3.

The following acceptance criteria are used in method validation:The systemic fault must not exceed its confidence interval (criteria of statistical insignificance);The confidence interval must include 100% of the extraction coefficient value.

It follows from the data in Table 3 that the procedure being validated has acceptable accuracy, as 100% is included in the confidence interval, and the bias is statistically indistinguishable from zero (0.63 ≤ 0.82).

### 2.5. Repeatability and Intra-laboratory Precision

Results of relation spiked-and-recovery (Z_i_) measurements obtained in repeatability and intra-laboratory precision conditions and their statistical processing are presented in Table 4.

Acceptance of intra-laboratory precision can be evaluated by the Fisher (*F*) and Student (*t*) statistical criteria by counting and comparing their actual *t* and *F* values with table values-maximal values of criteria with influence of random factors, current degrees of freedom, and given levels of significance. The data presented in Table 4, show a statistical insignificance of difference between means and standard deviations of two operators measures at a significance level of 95%, so the table *F* and *t* values substantially exceed their actual values.

### 2.6. Robustness

The study provided experimental evidence of the insensitivity of the procedure being validated to minor changes in the qNMR test conditions. Varying of the pulse angle, relaxation time, sample temperature, and addition of excipients do not change the position of the chemical shifts of GP and ImpA signals. The integral intensity ratio of GP and ImpA remains unchanged.

### 2.7. Comparative Analysis of the ImpA Content Determining by qNMR and HPLC

Results of the ImpA content in GP APIs and capsules determination, using qNMR and HPLC methods, are shown in Table 5.

It should be noted that content of ImpA in API test samples is lower than the LOQ of qNMR methods (0.025 w %) and HPLC methods (0.5 w %). The content level of lactam in capsules is more than the LOQ. ImpA content values in GP drugs, obtained by qNMR and HPLC methods, are close to each other, which is additional evidence of the accuracy of the validation method.

## 3. Materials and Methods

### 3.1. Materials

The following materials were used in the qNMR procedure development and validation: certified reference standards for GP and ImpA, manufactured by the European Pharmacopoeia (the assigned values of reference standards are 100%, the uncertainty of the assigned values is not stated), gabapentin APIs by Divis Laboratories Limited Hyderabad, India (A), gabapentin APIs by PIQ-PHARMA, Belgorod, Russia (B), gabapentin capsules by Canonpharma production PJSC, Moscow, Russia (I), and gabapentin capsules by Pharmstandard-Leksredstva JSC, Volginskiy, Russia (II). Deuterated dimethylsulfoxide (DMSO-D6, 99.90% D) and water (99.93% D) by Cambridge Isotope Laboratories, Inc. (St. Louis, MO, USA) were used in the NMR experiments.

HPLC measurements were carried out using ammonium dihydrogen phosphate, phosphoric acid, ACS grade perchloric acid, and sodium perchlorate (Sigma-Aldrich, St. Louis, MO, USA). ACS grade potassium dihydrogen phosphate was purchased from JT Baker (Philipsburg, NJ, USA). Methanol and HPLC grade acetonitrile were purchased from Fisher Scientific (Fairlawn, NJ, USA). HPLC ready 18 MΩ water was obtained, in-house, from a Milli-Q Integral 3 water purification system, Merck Millipore Corp. (Burlington, MA, USA). Syringe filters were used with PTFE membranes of 0.45 µm from Thermo Scientific Nalgene (Rochester, NY, USA).

### 3.2. NMR Spectroscopy Method

#### 3.2.1. Model Solutions

GP stock solution I of 100 mg/mL was prepared by placing 501.1 mg of GP reference standard in a 5-mL flask and diluting with D_2_O to volume. ImpA stock solution II (c = 2.026 mg/mL) was prepared by placing 10.13 mg of ImpA reference standard in a 5-mL flask and diluting with D_2_O to volume. Solution III (c = 506.5 μg/mL) was obtained by fourfold dilution of Solution II with D_2_O. Model solutions of GP and ImpA mixtures were prepared by combining different volumes of Solutions I and III and different volumes of solvents (Table 6). Trace amounts of DMSO-D6 were added as internal standards for the chemical shift scale calibration.

#### 3.2.2. Sample Preparation

API: 20 mg of substance (an accurate amount is optional) was placed into an NMR flask, followed by adding 0.5 mL of D_2_O and 10 µL of DMSO-D6, and shaking intensively to obtain a fully diluted sample.

Capsules: 1.5 mL of D_2_O was added to 1/2 of capsule content (200 mg, accurate amount is optional) and shaken intensively within 10 min. We obtained suspension filtered using a membrane filter, put 0.5 mL of filtrate in an NMR flask, and added 10 µL of DMSO-D6.

#### 3.2.3. Instrumentation and Experiments Conditions

^1^H spectra were collected on the Agilent DD2 NMR System 600 NMR spectrometer equipped with a 5 mm broadband probe and a gradient coil (VNMRJ 4.2 software). Parameters of the experiments: temperature 27 °C, spectral width 6009.6 Hz, observe pulse 90°, acquisition time 5.325 s, relaxation delay 10 s, number of scans 256, the number of analog-to-digital conversion points 64 K, exponential multiplication 0.3 Hz, zero filling 64 K, automatic linear correction of the baseline of the spectrum, manual phase adjustment, calibration of the δ scale under DMSO in D_2_O (δ = 2.71 ppm) [11]. The manual mode was also used for the signal integration. The general rule for choosing the integration limit (64 time the half-with of a Lorentzian shape NMR signal) was not followed due to the GP ^13^C satellites interfering effect. We took as the integration limit for ImpA the doubled distance between the center of its signal and GP ^13^C satellites. The integration limit for GP signal was equal to the distance between the^13^C satellites signals (without the ^13^C satellites). The relaxation delay value was estimated by an inversion-recovery experiment: T1 are equal 1.54 s (ImpA) and 0.89 s (GP). It was found that the experiment conditions did not affect the stability of GP: additional signals of ImpA were not detected in the spectrum of GP stock solution I.

### 3.3. Method Validation

Three independent experiments were run for each model solution and three values were obtained for the integral intensity of the signal. For validation, the mean value was used. Validation characteristics (specificity, linearity, accuracy, precision, limit of quantitation, range, and robustness) and validation criteria carried out according to methodological documents of GMP and US Pharmacopoeia guidance about validation of analytical procedure by qNMR [12,13]. Statistical parameters (mean value, standard deviation, coefficient of variation, significance interval, coefficient of determination, and actual and tabulated values for Fisher’s *F* test and Student’s *t* test) were determined at a significance level of *p* = 0.05 using MS Excel 2007.

#### 3.3.1. Specificity

Specificity was confirmed by demonstrating absence of overlap of individual GP and ImpA signals in the ^1^H spectrum.

#### 3.3.2. Limit of Quantitation

The limit of quantitation (LOQ) of the procedure was determined from the signal-to-noise ratio (S/N = 10) using VNMRJ software, version 4.2.

#### 3.3.3. Analytical Range

Range of the application method was determined by experimental value LOQ and the US Pharmacopoeia recommendation to nominal content of ImpA in GP APIs (0.1 w %) and in its marketed products (0.4 *w/w* %) [4,14,15].

#### 3.3.4. Linearity

Graphic dependence of the integral intensity of signal ImpA versus its concentration was treated by linear least square regression analysis with 10 model solutions over a concentration range of 0−253 µg/mL for ImpA.

#### 3.3.5. The Accuracy

The accuracy of the method was evaluated using the experimental data obtained from the linearity studies. Extraction coefficients were determined for all model solutions, i.e., the spiked-recovery ratio (Zi), for which the systematic error (δ), standard deviation (s), coefficient of variation (RSD), and significance interval (Δ) were determined.

#### 3.3.6. Precision

Precision was evaluated at the level of convergence and intralaboratory precision (different operators, different days). Convergence and intra-laboratory precision of the method under validation was evaluated using three model solutions with low (10 µg/mL), intermediate (100 µg/mL), and high (250 µg/mL) ImpA contents.

#### 3.3.7. Robustness

The reliability of an analytical measurement was evaluated by analyzing of the result stability after varying observe pulse (45 and 90°), relaxation delay (±10%), probe temperature (±2 °C), possible interfering species—water soluble excipients from marketed products (polyethylenglycol 6000).

### 3.4. Reference Measurement with HPLC Method

#### 3.4.1. Preparation of Solution

Diluent, a system suitability test solution, buffer solution, teste solution of samples A–D, reference solutions, and mobile phase were prepared according to USP methods [4,14].

#### 3.4.2. Instrumentation and Chromatographic Conditions

The HPLC system consists of an Agilent Infinity 1260 series (Agilent Technologies, Wilmington, DE, USA). Data collection and analysis were performed using ChemStation software. Chromatographic conditions: column Zorbax RX-C-18 250 mm × 4.6 mm × 5 µm (Agilent Technologies, Santa-Clara, CA, USA); column temperature 40 °C; elution mode isocratic; flow rate 1 mL/min; detector UV 215 nm; injection volume 20 μL; Run time no less than 50 min.

## 4. Conclusions

The developed ^1^H qNMR spectroscopy method of ImpA identification and quantification in GP APIs and GP drugs were validated by using the main parameters. It was established that the developed method is specific and has an acceptable linearity, repeatability, accuracy, precision, and robustness. Also, a limit of quantitation of developed method was established. This method can be used for carrying out GP APIs and drugs analysis.

## Figures and Tables

**Figure 1 molecules-26-01656-f001:**
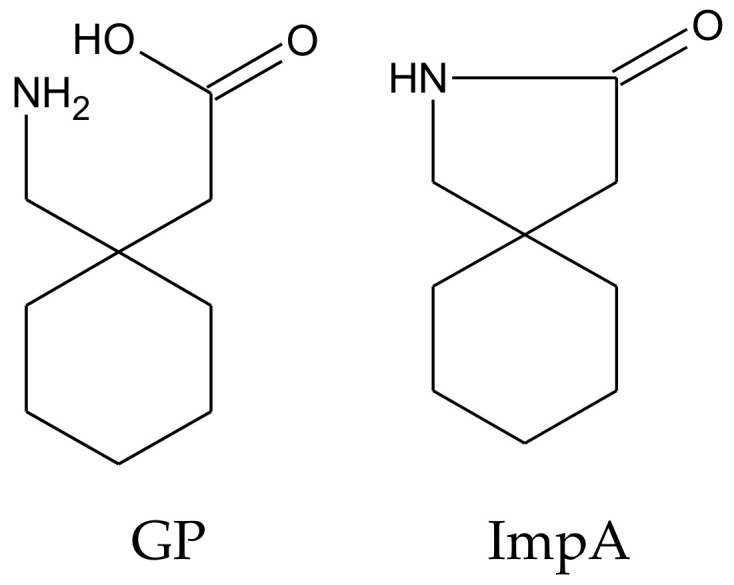
Chemical structures of Gabapentin (GP) and impurity A (impA).

**Figure 2 molecules-26-01656-f002:**
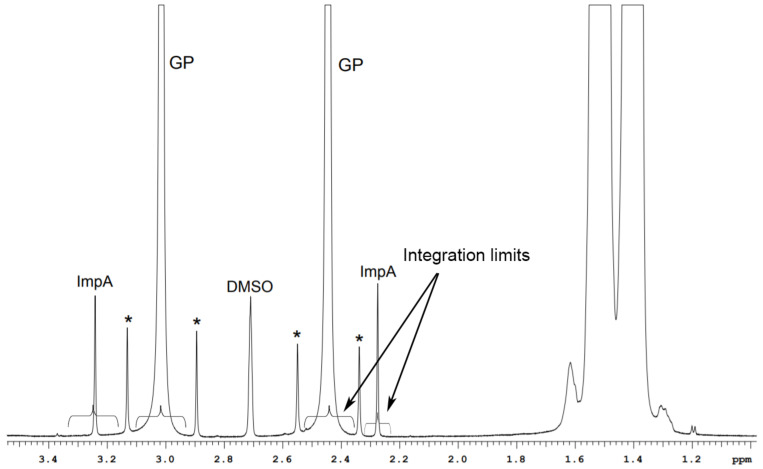
^1^H spectrum of GP and ImpA mixture (GP 40.09 mg/mL, ImpA 0.20 mg/mL). * ^13^C satellites of GP signals.

**Figure 3 molecules-26-01656-f003:**
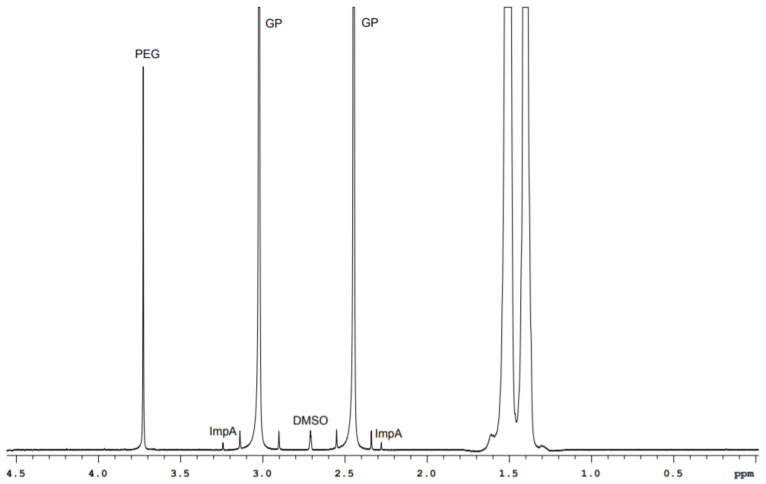
^1^H spectrum fragment of GP drug (capsules I).

**Figure 4 molecules-26-01656-f004:**
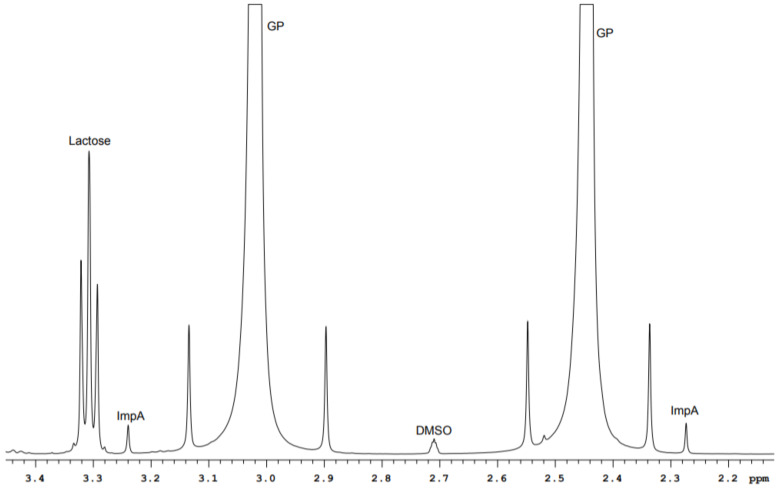
^1^H spectrum fragment of GP drug (capsules II).

**Figure 5 molecules-26-01656-f005:**
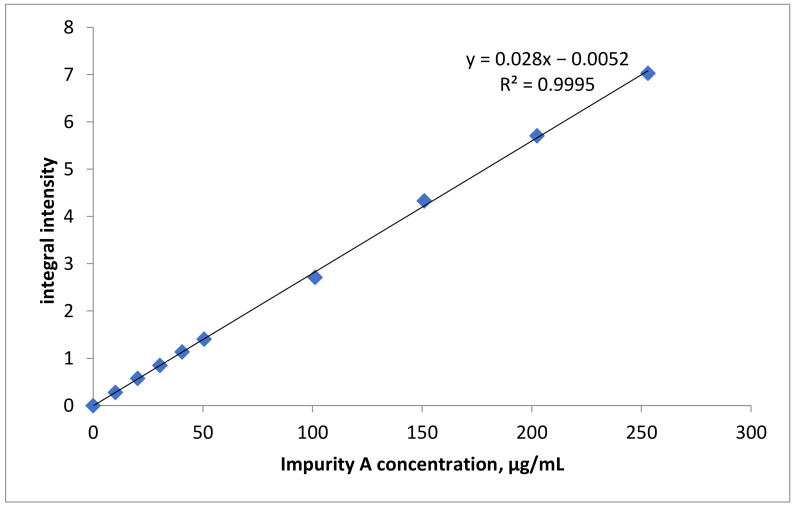
Dependence graph of ImpA measured integral signal intensity from its content in the sample.

**Table 1 molecules-26-01656-t001:** Results of the linearity evaluating of the validated method.

Content of ImpA, µg/mL (w % Relative to GP)	I_ImpA_	Mean Value I_ImpA_	Content of ImpA, µg/mL (w % Relative to GP)	I_ImpA_	Mean Value I_ImpA_
0.0 (0.0)	0.00	0.00	50.65 (0.126)	1.40 1.41 1.41	1.41
10.13 (0.025)	0.28 0.29 0.27	0.28	101.30 (0.253)	2.74 2.69 2.71	2.71
20.26 (0.051)	0.57 0.57 0.58	0.57	151.95 (0.379)	4.33 4.35 4.31	4.33
30.39 (0.076)	0.84 0.86 0.84	0.85	202.60 (0.505)	5.68 5.70 5.74	5.71
40.52 (0.101)	1.14 1.13 1.14	1.14	253.25 (0.632)	7.03 7.05 7.01	7.03

**Table 2 molecules-26-01656-t002:** Statistical Characteristics of Linear Regression.

Statistical Characteristic	Result
Slope (b)	0.028
Segment on ordinate (a)	−0.0052
Significance interval (p = 95%)	−0.06 ÷ 0.05
Correlation coefficient (r)	0.9997

**Table 3 molecules-26-01656-t003:** Results of the Accuracy Studies of the method.

ImpA Added, µg/mL	ImpA Found, µg/mL	Z_i_, %	ImpA Added, µg/mL	ImpA Found, µg/mL	Z_i_, %
10.13	10.11	99.80	101.30	98.94	97.67
	10.47	103.36		97.13	95.88
	9.75	96.25		97.85	96.59
20.26	20.58	101.58	151.95	156.35	102.90
	20.58	101.58		157.07	103.37
	20.94	103.36		155.63	102.42
30.39	30.33	99.80	202.60	205.09	101.23
	31.05	102.17		205.82	101.59
	30.33	99.80		207.26	102.30
40.52	41.16	101.58	253.25	253.84	100.23
	40.80	100.69		254.56	100.52
	41.16	101.58		253.12	99.95
50.65	50.55	99.80			
	50.91	100.51			
	50.91	100.51			
Mean ($\bar{Z}$), %				100.63	
Systematic error (δ), %				0.63	
Standard deviation (s), %				2.067	
Coefficient of variation (R.S.D.), %				1.86	
Significant interval (Δ), %				±0.82	

**Table 4 molecules-26-01656-t004:** Results of convergence and intralaboratory precision studies of the method being validated.

ImpA Added, µg/mL	Operator 1		Operator 2	
	Found, µg/mL	Z_i_, %	Found, µg/mL	Z_i_, %
10.13	10.11	99.80	10.11	99.80
	10.47	103.36	10.11	99.80
	9.75	96.25	9.39	92.69
50.65	50.55	99.80	51.27	101.22
	50.91	100.51	50.19	99.09
	50.91	100.51	50.91	100.51
253.25	253.84	100.23	253.48	100.09
	254.56	100.52	253.12	99.95
	253.12	99.95	253.84	100.23
Mean ($\bar{Z}$_i_), %		100,103	99.264	
Systematic error (δ), %		0.103	0.736	
Standard deviation (s), %		1.809	2.532	
Coefficient of variation (R.S.D.), %		1.807	2.551	
Significant interval, % (Δ, p = 95%)		±1.391	±1.946	
Combined mean ( $\bar{Z}$), %			99.684	
Combined standard deviation, %			2.20	
Combined coefficient of variation, %			2.207	
Combined significant interval, %			±1.555	
Fisher’s *F* test (F_tab_ = 3.44)			F_fact_ = 0.51	
Student’s *t* test (t_tab_ = 2.12)			t_fact_ = 0.81	

**Table 5 molecules-26-01656-t005:** Results of the ImpA content in GP APIs and capsules determination.

Sample	Content of ImpA, w %	
	NMR	HPLC
API A	Not found	Not found
API B	BQL	BQL
Capsule I	0.10 (RSD 5.6%)	0.13 (RSD 4.9%)
Capsule II	0.08 (RSD 7.5%)	0.07 (RSD 7.2%)

**Table 6 molecules-26-01656-t006:** Preparation of model solutions of GP and ImpA mixtures.

№	V I mL	V III µL	V DMSO-D6 µL	V D_2_O µL	C GP mg/mL	C ImpA µg/mL	w % ImpA Relative to GP
1	0.4	0	10	590	40.09	0	0
2	0.4	20	10	570	40.09	10.13	0.025
3	0.4	40	10	550	40.09	20.26	0.051
4	0.4	60	10	530	40.09	30.39	0.076
5	0.4	80	10	510	40.09	40.52	0.101
6	0.4	100	10	490	40.09	50.65	0.126
7	0.4	200	10	390	40.09	101.30	0.253
8	0.4	300	10	290	40.09	151.95	0.379
9	0.4	400	10	190	40.09	202.60	0.505
10	0.4	500	10	90	40.09	253.25	0.632

## Data Availability

The data that support the findings of this study are available from the corresponding author upon reasonable request.

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
