# Peer review of "Development and Validation of 2-Azaspiro [4,5] Decan-3-One (Impurity A) in Gabapentin Determination Method Using qNMR Spectroscopy"

_molecules, 2021, doi:10.3390/molecules26061656_

Round 1

Reviewer 1 Report

The authors developed a 1H-qNMR test method for quantitation of impurity A in gabapentin products, and verified the test method through the validation studies. The research process and results seem quite robust and systematic. Therefore, I think these results are worth publishing. However, it seems necessary to correct various minor errors during the manuscript writing and editing process.

  • First of all, I think it's probably problems of editing, but there are too many things that are incorrectly marked with "," instead of a decimal point in the expression of numbers.
  • In addition, there are many mislabeled units such as in concentration and weight, which require correction. (e.g. Table 6 and elsewhere)
  • It seems necessary to show the general qNMR equation before the impurity quantification equation following line 78.
  • Since LOD and LOQ are very important in a test method for quantifying impurity, a more detailed explanation is needed in Section "2.2.".
  • In Table 1, why are the significant figures of the impurity signal value and the mean value different?
  • Check the LOQ value given on line 84. (mkg/ml?)
  • Define "Zi" in Table 3.

Overall, there are too many typos and errors, so it seems necessary to review the manuscript very carefully

Author Response

Dear Reviewer! We are grateful for the careful analysis of the manuscript. All comments are taken into account.

  1. Tables 1-6 and everywhere in the text: The comma is replaced with a decimal point in the expression of numbers.
  2. Table 6 and everywhere in the text: units (concentration and weight) were adjusted.
  3. Page 3, line 80/83: the general qNMR equation is shown before the impurity quantification equation.
  4. Page 4, line 94: an explanation of the experimental LOQ value is given.
  5. Table 1: the significant figures of the impurity signal value and the mean value are aligned.
  6. Table 3: mean () value is given.

Reviewer 2 Report

Major questions and comments:

A repetition rate of 12.73 s (relaxation delay + acquisition time) seems rather short, even if it is probable, that both similar compounds show similar relaxation times. After 5 x T1 99.3% of the spin magnetization is relaxed, this is the minimum for qNMR. Have the relaxation times be estimated by an inversion-recovery experiment?

6000 Hz spectral with and 64K time domain points result in a digital resolution of 0.183 Hz/Pt. With a higher digital resolution, the precision could be increased even more. Probably a doubling of the acquisition time (with a corresponding reduction of the relaxation delay) would give even better results.
Have zero filling been used before Fourier transformation?

Has the stability of the analytes been tested under the analytical conditions? This seems to be particularly important since ImpA is formed by intramolecular cyclization of GP.

Fig. 2: Could you please specify the concentration of ImpA?

Page 4: "Integral intensity of signals δ 3,24 ppm 90 (ImpA) and 3,02 ppm (GP) were used in quantification measurements." Please mark in Fig. 2 the integration limits used (and increase the vertical scale). With an integration limit of 64 time the half-with of a Lorentzian shape NMR signal,  99% of the total intensity will be captured. This is not always practical, but for the traceability of the measurement, the integration limits should be specified.

Are the 13C satellites of GP included in the integration limits? As far as one can see from Fig.2, it is not possible to include the 13C satellites for both compounds, ImpA and GP. Thus, it might make more sense to exclude the satellites for both compounds. However, the integration limits should not be too small (Lorentzian line shape).  A reasonable compromise must be found. Can you please comment on this for your case?

Is the purity of the certified reference standards (GP and ImpA) known?

The quality of baseline and phase correction has a strong impact on the precision of the results. Which kind of "automatic correction of the baseline" has been used?

Minor remarks:

Keywords: Please replace "NMR" by "qNMR", or add "qNMR".

Please replace "satellites 13C" with "13C satellites".

Some Russian words/Cyrillic letters still need to be replaced

  • page 2, lines 60 & 61: "and" instead of "i"
  • Table 6: mL, mL (microliter), mg/mL, mg/mL (microgramm/L)

Some "mk" have to exchange by m (mikro set in Symbol): page 4, line 87; page 8, lines 171 and 174.

At least in the PDF format, formatting errors seem to appear in Tab. 4 - please check.

Page 7, line 128/129: Replace "angle rotation pulse magnetization" by "pulse angle".

Tab. 5 second row: Replace "HPLS" by "HPLC".

Page 8, line 177: Replace "multinuclear sensor" by "broadband probe".

Page 9, line 209: Replace "Etraction" by "Extraction"

3.4.2 Instrumentation and chromatographic conditions: Please add the elution solvent used.

Author Response

Dear Reviewer! We are grateful for the careful analysis of the manuscript. All comments are taken into account.

  1. Page 8, line 190: clarification about relaxation delay was added.
  2. Page 8, line 191: zero filling value was added.
  3. Page 9, line 200/202: clarification about stability of the gabapentin under the analytical conditions was added.
  4. Fig. 2: in the caption to the picture, the concentrations of ImpA and GP were indicated.
  5. Fig. 2: the vertical scale was increased, integration limits were marked.
  6. Page 9, line 194/199: the integration limits should be specified.
  7. Page 7, line 155/156: information about the purity of the certified reference standards was added.
  8. Page 8, line 192: the kind of automatic correction of the baseline was added (linear).
  9. Everywhere in the text: "NMR" was replaced by "qNMR".
  10. Everywhere in the text: "satellites 13C" was replaced by "13C satellites".
  11. Page 2, lines 61 & 62, table 6: some russian words/cyrillic letters have been replaced.

12.Everywhere in the text: some "mk" have been exchanged by µ (mikro set in Symbol).

  1. Page 7, line 138: Replace "angle rotation pulse magnetization" have been replaced by "pulse angle".
  2. Tab. 5 second row: "HPLS" have been replaced by "HPLC".
  3. Page 8, line 188: "multinuclear sensor" have been replaced by "broadband probe".
  4. Page 9, line 229: "Etraction" have been replaced by "Extraction".
  5. Page 10, line 244: information about diluent was added.
